

# Genetic diversity and its conservation implications of *Vitex rotundifolia* (Lamiaceae) populations in East Asia

Yiqi Sun[1,*], Hong Yang[2,*], Qiaoyan Zhang[8], Luping Qin[3,8], Pan Li[4], Joongku Lee[5], Shichao Chen[6], Khalid Rahman[7], Tingguo Kang[1] and Min Jia[3]

[1] College of Pharmacy, Liaoning University of Traditional Chinese Medicine, Dalian, Dalian, Liaoning, China
[2] School of Medicine, Tongji University, Shanghai, China
[3] Department of Pharmacognosy, Second Military Medical University School of Pharmacy, Shanghai, China
[4] Key Laboratory of Conservation Biology for Endangered Wildlife of the Ministry of Education, and Laboratory of Systematic & Evolutionary Botany and Biodiversity, College of Life Sciences, Zhejiang University, Hangzhou, Zhejiang, China
[5] Department of Environment and Forest Resources, College of Agriculture & Life Sciences, Chungnam National University, Daejeon, South Korea
[6] College of Life Science and Technology, Tongji University, Shanghai, China
[7] School of Pharmacy and Biomolecular Sciences, Liverpool John Moores University, Liverpool, United Kingdom
[8] College of Pharmaceutical Sciences, Zhejiang Chinese Medical University, Hangzhou, Zhejiang, China
* These authors contributed equally to this work.

Corresponding authors
Tingguo Kang, kangtingguo@163.com
Min Jia, jm7.1@163.com

## ABSTRACT

*Vitex rotundifolia* is an important coastal and medicinal plant, and is recorded in the List of the Important Wild Plants for Conservation in China and Japan. However, an effective conservation strategy is lacking. In the present study, the genetic diversity and population structure were analyzed using phylogeographical methods based on the *trnH-psbA* and *trnG-trnS* intergenic spacers of the chloroplast DNA (cpDNA) sequences from 157 individuals from 25 sampling sites for *V. rotundifolia* and *V. trifolia* plus the internal transcribed spacer (ITS) of the nuclear ribosomal DNA (nrDNA) sequences of 177 individuals from 27 sampling sites. The results showed that *V. rotundifolia* and *V. trifolia* had eight cpDNA and two nrDNA haplotypes, respectively, and the *V. rotundifolia* has a low level of genetic diversity (haplotype diversity $h_{d,cp}$ = 0.360, $h_{d,nr}$ = 0.440), a more pronounced genetic differentiation among populations (population differentiation at the species level ($G_{ST}$) = 0.201, population differentiation at the allele level ($N_{ST}$) = 0.462), and an insignificantly different phylogeographical structure ($N_{ST} > G_{ST}$, $P > 0.05$). In addition, haplotype network analyses indicated that *V. rotundifolia* and *V. trifolia* have distinct haplotypes. Divergence dating based on BEAST software analyses showed that most cpDNA clades diverged in the late Pleistocene era. Demographic analysis indicated that *V. rotundifolia* underwent a rapid demographic expansion. Some scientific strategies are suggested for resource conservation of *V. rotundifolia* based on its genetic diversity and population structure.

## INTRODUCTION

*Vitex rotundifolia*, which often grows on beaches and sand dunes, is a widely distributed shoreline shrub of the family Lamiaceae, (*Cantino, 1992*; *Harley et al., 2004*; *Cousins et al., 2010*). *V. rotundifolia* plays an important ecological role in stabilizing sand dunes in coastal areas (*Gresham & Neal, 2004*; *Kim, 2005*). In addition, the fruit of *V. rotundifolia*, known as Manjingzi, is a herbal medicine commonly used in China and Japan to prevent and treat colds, headache, and migraine (*Sung et al., 1996*). However, the increased use of the plant as a medical resource, coastal overexploitation, and environmental destruction, have accelerated the degradation of *V. rotundifolia* populations, decreased the intraspecific variation, severely destroyed its natural habitats, and even threatened the survival of the species. Given the vulnerability of *V. rotundifolia*, it has been recorded in the List of the Important Wild Plants for Conservation in China and Japan (*Endangered Species Scientific Commission, P.R.C., 2018*; *Ohtsuki et al., 2014*). Therefore, it is necessary to provide an effective conservation strategy for this species.

Based on a comprehensive allozyme study of *V. rotundifolia* Korean populations, *Yeeh et al. (1996)* reported that the levels of genetic variation and differentiation within populations are considerably lower, but are higher among populations. By contrast, levels of genotypic diversity within and among populations were moderate. They indicated that clonal reproduction might act as an enhancer of genetic drift by reducing the effective size of local *V. rotundifolia* populations. *Ohtsuki et al. (2014)* developed ten microsatellite makers from *V. rotundifolia* Japanese populations; however, no further data were presented. *Hu et al. (2008)* investigated the genetic variation of *V. rotundifolia* at more extensive sampling areas in Chinese populations using inter simple sequence repeat (ISSR) markers. Their fine-scale spatial autocorrelation analysis showed a clear within-population structure, with gene clusters of approximately 20 m, which could act as a guide for sampling strategies. They also reported the overall genetic diversity (GD) of *V. rotundifolia* on China was moderate (GD = 0.190) and the genotypic diversity was greater than the average values for a clonal plant, indicating its significant reproduction through seedlings. Pharmacological research discovered that the genetic variation pattern was closely associated with the chemical constituents in the fruit of *V. rotundifolia* (*Hu et al., 2007*). Currently, our understanding of the genetic variation background of this species is poor, especially concerning the population genetic structure and phylogeography. In addition, we lack specific advice for the conservation of *V. rotundifolia*.

*V. trifolia* is another important medicinal plant and is the closest relative of *V. rotundifolia*. *V. trifolia* is only distributed in a very narrow area in Southwest of China and Taiwan province, and the independent species taxonomic status between *V. trifolia* and *V. rotundifolia* has long been disputed (*Munir, 1987*; *Kok, 2007*; *Kok, 2008*; *Barger et al., 2012*; *Chang, Kim & Chang, 2014*; *Franck et al., 2016*). The fruit of *V. trifolia* have also been recorded in the Pharmacopoeia of China as Manjingzi (*Editorial Committee of Chinese Pharmacopoeia, 2015*) for medical use. Therefore, the addition of *V. trifolia* in this study could help to analyze the evolutionary history of *V. rotundifolia* and to better understand their genetic differentiation.

The analyses of population genetic structures, genetic variations across populations, and the geographical distributions of the species at risk of extinction will help to develop appropriate conservation decisions and sustainable utilization strategies (*Millar & Libby, 1991*). Phylogeographical methods are often used to assess the variability of molecular markers within a taxon across both time and space, to deduce the historical processes that may have been responsible for the contemporary geographical distributions of individuals and population genetic structures, and help to prioritize areas of high value for conservation (*Avise, 2000*; *Franck et al., 2016*; *Kumar & Kumar, 2018*). Chloroplast DNA (cpDNA), which is maternally inherited in most angiosperm plants, is commonly considered as a single non-recombinant unit of inheritance and often is used to investigate the phylogeographical processes associated with seed dispersal, such as range expansion (*Clegg et al., 1994*; *Petit et al., 1997*). Nuclear ribosomal DNA (nrDNA), which exhibits different modes of inheritance, can also be used to determine a species' genetic structure. The combination of highly conserved and variable regions allows us to make phylogenetic inferences across a wide range of evolutionary timescales and to unravel the genetic variation and demographic history of a plant species (*Weitemier et al., 2015*).

In the present study, we analyzed the cpDNA and internal transcribed spacer (ITS) haplotype sequences from 23 sampling sites covering most distribution ranges of *V. rotundifolia* in East Asia; and four sampling sites of *V. trifolia*, to explore the genetic variation pattern, population structure, and the evolutionary history of *V. rotundifolia*, with the aim of proposing effective strategies for its resource conservation.

## MATERIALS AND METHODS

### Population sampling

A total of 157 individuals of *V. rotundifolia* and *V. trifolia* from 25 sampling sites for cpDNA and 177 individuals of *V. rotundifolia* and *V. trifolia* from 27 sampling sites for ITS were sampled from distribution areas in East Asia (Table 1 and Fig. 1). Leaf samples were collected from individuals separated by more than 20 m to avoid collecting the same clones (genotypes). The leaf samples were dried in silica gel and then stored in a freezer ($-20\,°C$) for use. A sample of *V. negundo* from Shanghai was also collected as the outgroup. Voucher specimens were deposited in the herbarium of the Second Military Medical University (Shanghai, China).

### Sequencing of cpDNA and nrDNA

The noncoding regions DNA sequences of cpDNA *trnG* (encoding tRNA glycine)-*trnS* (encoding tRNA serine) and *trnH* (encoding tRNA Histidine)-*psbA* (encoding photosystem II protein D1) and ITS intergenic spacer region DNA sequences were examined to analyze the genetic variation in *V. rotundifolia* and *V. trifolia* from different geographical locations (*Hamilton, 1999*; *Shaw et al., 2007*; *Zhang et al., 2014*). Total DNA was extracted using a Hi-DNA secure Plant Kit (Tiangen Biotech (Beijing) Co., Ltd, Beijing, China). PCR amplification was conducted in a total reaction volume of 50 µl, containing 10–20 ng total DNA, with 1 µl of each primer, 21 µl of ddH$_2$O, and 25 µl of 2× Trans Taq High Fidelity (HiFi) PCR SuperMix (Transgen Biotech, Beijing, China). Double-stranded DNA

Sun et al. (2019), *PeerJ*, DOI 10.7717/peerj.6194

**Table 1** Localities and numbers of samples surveyed across the natural range of *V. rotundifolia* and *V. trifolia* and the numbers of haplotypes of cpDNA and nrDNA observed in the populations surveyed.

| Population | | Latitude (N) | Longitude (E) | Altitude (m) | $n_{s,cp}/n_{s,nr}$ | $n_{h,cp}/n_{h,nr}$ | $h_{cp}/h_{nr}$ | $\pi_{cp}/\pi_{nr}$ |
|---|---|---|---|---|---|---|---|---|
| *Vitex. rotundifolia* | | | | | | | | |
| DL | Dalian, Liaoning, China | 39°10′11.62″ | 22°04′40.26″ | 28 | 10/11 | 2/2 | 0.467/0.356 | 0.00042/0.00106 |
| WH | Weihai, Shandong, China | 37°31′60.75″ | 22°02′.25.55″ | 6 | 9/8 | 2/2 | 0.389/0.429 | 0.00035/0.00128 |
| QD | Qingdao, Shandong, China | 36°24′78.35″ | 20°55′48.07″ | 5 | 9/10 | 2/2 | 0.500/0.467 | 0.00045/0.00140 |
| LYG | Lianyungang, Jiangsu, China | 34°45′74.87″ | 9°28′37.83″ | 1 | 4/7 | 1/2 | 0.000/0.571 | 0.00000/0.00171 |
| NJ | Nanjing, Jiangsu, China | 32°03′20.78″ | 118°49′19.85″ | 25 | 5/5 | 1/2 | 0.000/0.600 | 0.00000/0.00180 |
| DC | Duchang, Jiangxi, China | 29°26′06.09″ | 116°06′42.05″ | 76 | 11/12 | 2/2 | 0.436/0.409 | 0.00039/0.00123 |
| HT | Houtian, Jiangxi, China | 28°26′20.26″ | 115°48′00.26″ | 39 | 12/12 | 1/1 | 0.000/0.000 | 0.00000/0.00000 |
| ZS | Zhoushan, Zhejiang, China | 29°52′56.42″ | 22°24′.02.03″ | 0 | 9/10 | 1/2 | 0.000/0.200 | 0.00000/0.00060 |
| XS | Xiangshan, Zhejiang, China | 29°25′86.14″ | 21°57′76.63″ | 3 | 2/3 | 1/2 | 0.000/0.667 | 0.00000/0.00200 |
| TZ | Taizhou, Zhejiang, China | 28°41′46.83″ | 21°46′54.73″ | 10 | 9/12 | 2/2 | 0.389/0.530 | 0.00035/0.00159 |
| SH | Shanghai, China | 31°13′55.03″ | 121°28′10.00″ | 13.31 | 4/4 | 1/2 | 0.000/0.667 | 0.00000/0.00200 |
| XM | Xiamen, Fujian, China | 24°28′56.93″ | 118°05′4.03″ | 2 | 1/1 | 1/1 | 0.000/0.000 | 0.00000/0.00000 |
| GD | Gangzhou, Guangdong, China | 23°11′21.66″ | 113°21′31.41″ | 22 | 11/10 | 2/2 | 0.467/0.467 | 0.00042/0.00140 |
| 1_WC | Wenchang, Hainnan, China | 19°32′42.56″ | 110°47′51.87″ | 10 | 8/10 | 1/2 | 0.000/0.467 | 0.00000/0.00140 |
| WN | Wanning, Hainnan, China | 18°47′46.89″ | 110°23′27.86″ | 13 | 9/11 | 3/2 | 0.639/0.436 | 0.00065/0.00131 |
| XL | Xinglong, Hainnan, China | 18°44′4.68″ | 110°11′52.91″ | 26 | 10/10 | 2/2 | 0.533/0.200 | 0.00048/0.00060 |
| TW | Pingdong, Taiwan, China | 21°59′23.29″ | 120°44′21.12″ | 5 | 0/2 | 0/1 | 0/0.000 | 0/0.00000 |
| JP | Akita, Japan | 39°51′26.24″ | 140°0′46.99″ | 6.5 | 1/3 | 1/1 | 0.000/0.667 | 0.00000/0.00200 |
| JPYD | Lzu, shizuoka, Japan | 34°55′55.36″ | 139°07′32.80″ | 162 | 1/1 | 1/1 | 0.000/0.000 | 0.00000/0.00000 |
| KR1 | Boryeong, Jangan-beach, Korea | 36°12′49.9″ | 126°32′09.3″ | 16 | 0/1 | 0/1 | 0/0.000 | 0/0.00000 |
| KR2 | Jeollanam-do Hampyeong Hampyeong port, Korea | 35°09′30.4″ | 126°22′35.9″ | 12 | 1/1 | 1/1 | 0.000/0.000 | 0.00000/0.00000 |
| KR3 | Jeju-si, Iho-beach, Korea | 32°29′54.2″ | 126°27′14.7″ | 3.8 | 2/2 | 1/1 | 0.000/0.000 | 0.00000/0.00000 |
| JPCS | Nago, Okinawa, Japan | 26°35′29.57″ | 127°58′38.34″ | 5.8 | 3/3 | 1/2 | 0.000/0.000 | 0.00000/0.00200 |
| Species level | | | | | | | 0.360/0.440 | 0.00010/0.00000 |
| *Vitex. trifolia* | | | | | | | | |
| 3_GD | Gangzhou, Guangdong, China | 23°11′21.66″ | 113°21′31.41″ | 22 | 1/1 | 1/1 | 0.000/0.000 | 0.00000/0.00000 |
| 3_YN | Xishuangbanna, Yunnan, China | 22°0′38.81″ | 100°47′49.74″ | 555 | 9/8 | 3/2 | 0.639/0.571 | 0.00065/0.00171 |
| 3_WC | Wenchang, Hainnan, China | 19°32′42.56″ | 110°47′51.87″ | 10 | 10/11 | 3/2 | 0.511/0.545 | 0.00102/0.00163 |
| 3_GX | Longgang, Guangxi, China | 22°28′15.35″ | 106°57′25.94″ | 239 | 7/8 | 2/2 | 0.571/0.536 | 0.00051/0.00160 |
| Species level | | | | | | | 0.812/0.516 | 0.00039/0.00000 |

**Notes.**

$n_s$, the number of samples analysed; $n_h$, the number of haplotypes observeed; $h$, haplotype diversity; $\pi$, nucleotide diversity.
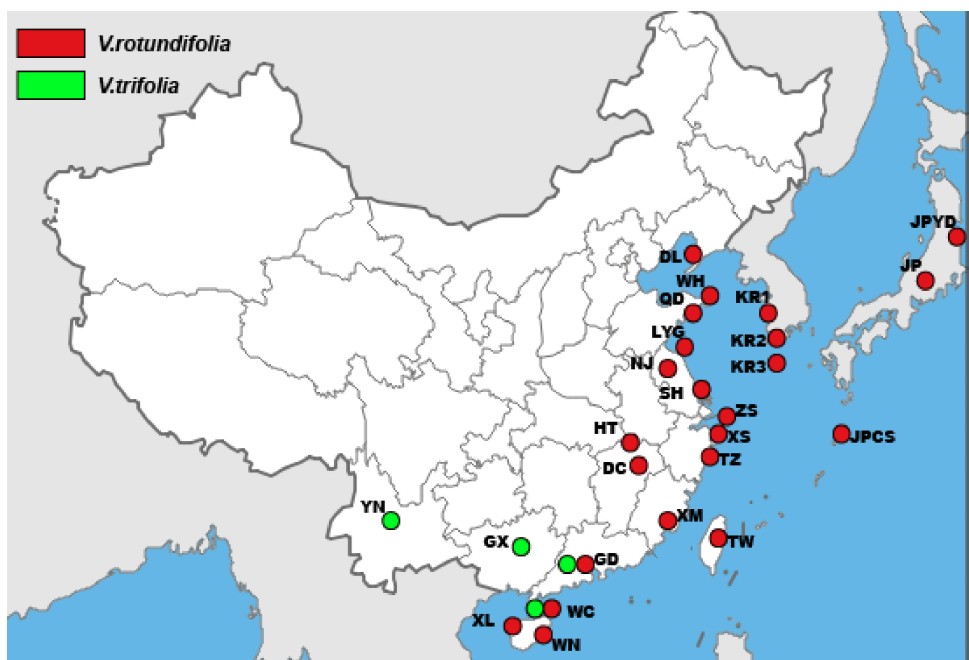

**Figure 1   The sample locations of *V. rotundifolia* and *V. trifolia* were used in this study.**

was amplified after 3-min incubation at 94 °C; followed by 30 cycles at 94 °C for 30 s, 54/56 °C for 30 s, and 72 °C for 30 s; with a final extension at 72 °C for 15 min. To examine the geographical distribution of cpDNA and nrDNA in *V. rotundifolia* and *V. trifolia*, we amplified two noncoding regions of cpDNA *trnG-trnS* and *trnH-psbA* and ITS intergenic spacer regions using primers that successfully amplified the expected DNA fragments and exhibited some variations in our preliminary experiments (*Hamilton, 1999*; *Shaw et al., 2007*; *Zhang et al., 2014*). Primers used for PCR amplification were also used as sequencing primers. The PCR products were assessed using electrophoresis through 1.0% agarose gels and then used as templates for direct sequencing. Sequencing was conducted from both ends. The cpDNA and ITS sequences data of *V. rotundifolia* and *V. trifolia* samples have been submitted to the GenBank database, and the accession numbers for the *trnG-trnS*, *trnH-psbA*, and ITS sequences are MG822129–MG822287, MG822468–MG822626, and MG822288–MG822467, respectively.

**Phylogenetic analyses**

Geneious Pro V4.8.5 (*Drummond et al., 2017*) was used to check and align the DNA sequences. Bayesian analysis (BI) was performed on MrBayes3.1.2 to construct the phylogenetic tree of haplotypes based on the combined cpDNA data (*Ronquist et al., 2012*). T92+G was selected as the model for the cpDNA dataset using Modeltest version 3.06 (*Posada & Crandall, 1998*) and the Akaike Information Criterion (AIC) was used to calculate the parameters and assumptions for the sequence partitions. Two runs of four simultaneous Monte Carlo Markov Chains (MCMC) analyses were performed for 10,000,000 generations, with sampling done every 100 generations. Log-likelihood values

were examined for stationarity to determine the burn-in value, which was 5,000 trees in this case. TCS 1.21 (*Clement, Posada & Crandall, 2000*) was applied to build a genotype network based on the 90% parsimony criteria with the option of treating gaps as a fifth base.

## Molecular dating

The program BEAST version 1.5.3 (*Drummond et al., 2012*) was used to estimate the divergence times of the major lineages to the most recent common ancestor (TMRCA). Considering there are no fossil records and no specific substitution rates to calibrate the molecular clocks, the published nucleotide substitution rates were used to estimate the divergence time between two species, and a rate of about $1.2–1.7 \times 10^{-9}$ substitutions per neutral site per year (s/s/y) was used to obtain absolute values of TMRCA (*Eyre-Walker, 2000*).

## Population genetic diversity and genetic differentiation

The program DnaSP v5.10 was used to calculate the haplotype diversity ($h_d$) and nucleotide diversity ($\pi_d$) based on the cpDNA and nrDNA sequences of *V. rotundifolia* and *V. trifolia* (*Librado & Rozas, 2009*). To prevent an insufficient sampling affect, we re-analyzed the nucleotide diversity of *V. rotundifolia* excluding poorly sampled populations.

The analyses of molecular variance (AMOVA) in ARLEQUIN 3.5 were performed to calculate the genetic variation between *V. rotundifolia* and *V. trifolia* groups, among populations within groups, and within populations, using a significance test based on 1,000 permutations (*Excoffier & Lischer, 2010*). The program PermutCpSSR_1.2.1 (*Pons & Petit, 1996*) was used to calculate the within-population diversity ($h_S$), total diversity ($h_T$), geographical total haplotype diversity ($V_T$), geographical average haplotype diversity ($V_S$), the level of population differentiation at the species level ($G_{ST}$), and an estimate of population subdivisions for phylogenetically ordered alleles ($N_{ST}$). $G_{ST}$ and $N_{ST}$ are often used to assess the geographical structure affecting population differentiation.

## Demographic history

Tajima's $D$ test and Fu's $F$s test in the ARLEQUIN 3.5 software, with 1,000 permutations, were applied to detect historical demographic range expansions of *V. rotundifolia* and *V. trifolia* (*Excoffier & Lischer, 2010*). The significance of the D value is associated with bottlenecks, selective effects, population expansion, or heterogeneity of mutation rates (*Tanaka et al., 2011*). In addition, 'mismatch distributions' analysis was performed using the DnaSP v5.10 program to detect any recent demographic expansions of *V. rotundifolia* or *V. trifolia* (*Librado & Rozas, 2009*).

# RESULTS

## Genetic diversity based on cpDNA and ITS sequences

Based on the 157 samples of *V. rotundifolia* and *V. trifolia* individuals, the nucleotide sequence lengths were 848 bp and 271 bp for the *trnG-trnS* and *trnH-psbA* regions, respectively. The combined cpDNA sequence length after multiple alignments was 1,119 bp. In total, six polymorphic sites and eight haplotypes were detected, and the aligned sequence

Peerj

**Table 2  Variable nucleotide sites and length polymorphisms of cpDNA (*trnH-psbA* and *trnG-trnS*) sequences in the *V. rotundifolia* and *V. trifolia*, identifying 9 haplotypes (Hap1-Hap9).**

| Population | Haplotype | cpDNA | | | | | | | | | | | | | | | | | | | |
|---|---|---|---|---|---|---|---|---|---|---|---|---|---|---|---|---|---|---|---|---|---|
| | | 407 | 672 | 728 | 782 | 789 | 790 | 797 | 802 | 828 | 848 | 867 | 882 | 883 | 891 | 892 | 893 | 996 | 1008 | 1041 | 1074 |
| 1_WC, DC, DL, GD, HT, JPCS, ZS, JPYD, KR3, LYG, NJ, QD, SH, TZ, WH, WN, XL, XS | Hap1 | A | - | G | - | T | C | T | - | - | C | T | - | - | T | T | T | G | G | G. | C |
| 3_GD, 3_GX, 3_WC | Hap2 | . | - | . | . | G | A | . | . | . | . | . | T | A | . | . | . | . | . | . | - |
| 3_GX, 3_WC | Hap3 | . | - | . | . | G | A | . | . | . | . | . | T | A | . | . | . | . | . | . | . |
| 3_WC, 3_YN | Hap4 | . | - | . | . | G | A | . | . | . | . | . | . | . | . | . | . | . | . | . | . |
| 3_YN | Hap5 | C | - | . | . | G | A | . | . | . | . | . | . | . | . | . | . | . | . | . | - |
| 3_YN | Hap6 | C | - | . | . | G | A | . | . | . | . | . | . | . | . | . | . | . | . | . | . |
| DC, DL, GD, KR2, QD, TZ, WH, WN | Hap7 | . | - | . | . | . | . | . | . | . | . | . | . | . | . | . | . | . | . | . | - |
| WN, XL, XM | Hap8 | C | - | . | . | . | . | . | . | . | . | . | . | . | . | . | . | . | . | . | . |
| Outgroup | Hap9 | . | A | A | T | G | A | A | T | G | A | A | . | . | A | C | A | T | A | C | . |

**Notes.**
All sequences are compared to the reference Hap1. "-" in sequences denote absence.

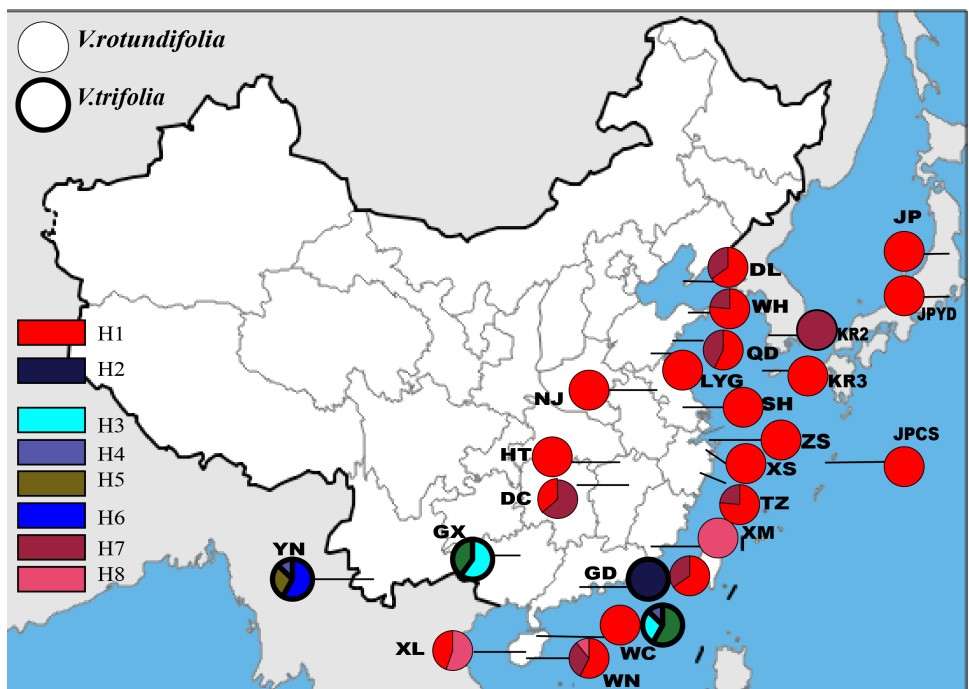

**Figure 2  Geographic distribution of the cpDNA haplotypes in *V. rotundifolia* and *V. trifolia*.**

variations are summarized in Table 2. The number of haplotypes, haplotype diversity ($h_d$), and the nucleotide diversity ($\pi_d$) within each population are presented in Table 1. Finally, three haplotypes were obtained from 17 populations, plus four sites with a single sample of *V. rotundifolia*, and five haplotypes were obtained from four sampling sites of *V. trifolia* (Fig. 2). The value of haplotype diversity ($h_{d,cp}$) ranged from 0 to 0.639, and the nucleotide diversity ($\pi_{d,cp}$) from 0 to 0.00065 in the 21 sampling sites for *V. rotundifolia*. At the species level, $h_{d,cp}$ and $\pi_{d,cp}$ were 0.360 and 0.00010, respectively. When excluding six poorly sampled populations, $h_{d,cp}$ and $\pi_{d,cp}$ slightly changed to 0.363 and 0.00010, respectively. Within the four *V. trifolia* sampling sites, $h_{d,cp}$ ranged from 0 to 0.639 and $\pi_{d,cp}$ from 0 to 0.00065, and at the species level, $h_{d,cp}$ was 0.812 and $\pi_{d,cp}$ was 0.00039. These results indicated that both *V. rotundifolia* and *V. trifolia* have relatively low genetic diversity. The highest haplotype diversity occurred in population WN for *V. rotundifolia* and 3_YN for *V. trifolia*, indicating that Wanning of Hainan Province and Xishuangbanna of Yunnan Province may be the centers of biological diversity of *V. rotundifolia* and *V. trifolia*, respectively.

The length of the ITS sequence was 334 bp for 177 individuals, but only two haplotypes and one polymorphic site were found in *V. rotundifolia* and *V. trifolia* (Fig. 3, Table 3). In 23 sampling sites for *V. rotundifolia*, the total haplotype diversity ($h_{d,nr}$) was 0.440, the range of haplotype diversities within population was from 0–0.667; the nucleotide diversity ($\pi_{d,nr}$) was 0.00000; and the range of nucleotide diversities within population was 0–0.00200. When we excluded six poorly sampled populations, the ITS total $h_{d,nr}$ and $\pi_{d,nr}$

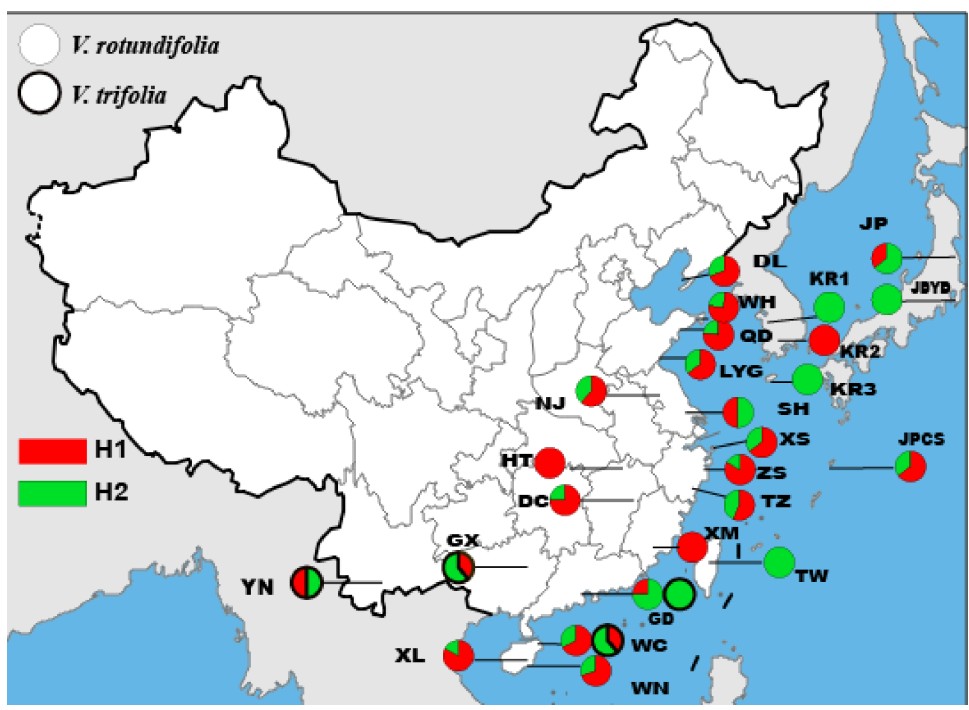

**Figure 3** Geographic distribution of the nrDNA haplotypes in *V. rotundifolia* and *V. trifolia.*

**Table 3** Variable nucleotide sites and length polymorphisms of nrDNA sequences in the *V. rotundifolia* and *V. trifolia*, identifying 3 haplotypes (Hap1-Hap3).

| Population | Haplotype | cpDNA | | | | | |
|---|---|---|---|---|---|---|---|
| | | 70 | 72 | 217 | 264 | 274 | 276 |
| 1_WC, 3_YN, 3_GX, 3_WC, DC, DL,r GD, HT, JP, JPCS, KR2, LYG, NJ, QD, SH, TZ, ZS, WH, WN, XL, XM,r XS, ZS | Hap1 | G | C | C | – | G | C |
| 1_WC, 3_YN, 3_GX, 3_WC, 3_GD, DC, DL, GD, HT, JP, JPCS, KR2, LYG, NJ, QD, SH, TZ, ZS, WH, WN, XL, XM, XS, ZS, KR1, KR3 | Hap2 | . | . | . | C | . | . |
| Outgroup | Hap3 | A | T | T | C | A | T |

**Notes.**
All sequences are compared to the reference Hap1. "-" in sequences denote absence.

slightly changed to 0.443 and 0.00000, respectively. Within the four *V. trifolia* sampling sites, the total values of $h_{d,nr}$ and $\pi_{d,nr}$ were 0.516 and 0.00000, respectively. These results indicated that based on the ITS nuclear sequence, *V. rotundifolia* and *V. trifolia* both have a low genetic diversity.

Furthermore, PermutCpSSR_1.2.1 analysis based on cpDNA sequences deduced that the $h_T$, $V_T$, $h_S$, and $V_S$ values for *V. rotundifolia* were 0.281, 0.285, 0.225, and 0.285, respectively, and those for the *V. trifolia* populations were 0.869, 0.921, 0.690, and 0.553, respectively,
indicating that *V. rotundifolia* populations also have low genetic diversity, and that the genetic diversity of the *V. trifolia* populations was much higher than that of *V. rotundifolia* populations. The parameters of $h_T$, $V_T$, $h_S$, and $V_S$ based on ITS sequence could not be calculated because only two haplotypes were found in *V. rotundifolia* and *V. trifolia*.

AMOVA analysis based on the cpDNA sequences revealed that a total of 79.19% of the variation was between the *V. rotundifolia* and *V. trifolia* populations, and 80.36% of the total cpDNA variation existed within *V. rotundifolia* populations. In addition, 78.25% of the variations based on cpDNA were found between *V. rotundifolia* and *V. trifolia* populations ($P < 0.01$; Table 3), indicating that there was a significant genetic differentiation between these two species. Based on ITS sequences from combined *V. rotundifolia* and *V. trifolia* populations, AMOVA analysis showed that 6.29% of the variation existed between the species, 6.70% of the variation was caused by differences between the populations within the species, and 87.02% of the variation existed within the populations (Table 4).

## Population structure and phylogeographical analysis

The phylogeographical analysis based on the cpDNA sequences found that the $N_{ST}$ value was significantly greater than the corresponding $G_{ST}$ value (0.751 *vs.* 0.436, $P < 0.05$), showing a noticeable phylogeographical structure across 25 *V. rotundifolia* and *V. trifolia* populations, and the eight haplotypes displayed a clear geographical pattern. However, the $N_{ST}$ value was higher than its corresponding $G_{ST}$ value (0.462 *vs.* 0.201, $P > 0.05$) in 21 *V. rotundifolia* populations, indicating that there was an unclear cpDNA phylogeographical structure. As shown in Fig. 4, statistical parsimony analysis found a single haplotype network based on cpDNA samples of *V. rotundifolia* and *V. trifolia* populations. Furthermore, the TCS 1.21 network analysis of cpDNA haplotypes revealed the relationship of the interior (ancestral) and the tip (derived) haplotypes. Haplotype H4 was inferred as the ancestral haplotype, as determined by outgroup weight based on haplotype positions in the network. H1, H7, and, H8 were shared only by *V. rotundifolia*. H2–H6 were shared only by *V. trifolia*. H1 was the most frequent and widely distributed haplotype in 17 of the 21 populations of *V. rotundifolia*.

The phylogenetic relationships between the eight cpDNA haplotypes were assessed under Bayesian inferences drawn using *V. negundo* as an outgroup. The result showed that the phylogenetic tree of the eight cpDNA haplotypes had a comb-like structure (Fig. 5). This result may have been caused by insufficient information sites, or by the rapid expansion of *V. rotundifolia* and *V. trifolia*. We could not perform the haplotype network and phylogenetic analyses based on the ITS sequences because there were only two haplotypes.

## Divergence times and demographic history

BEAST, a program for Bayesian analysis, was used to estimate the divergence times. The result showed that the effective sample size (ESS) value ranged from 919 to 1,070 for all the nodes, the divergence time between *V. rotundifolia* and *V. trifolia* was 0.23 mya (0.096–0.396, 95% highest posterior density (HPD)) (Fig. 6). Hence, the divergence between *V. rotundifolia* and *V. trifolia* appeared during the late Pleistocene era.

**Table 4  Analyses of molecular variance (AMOVA) based on the cpDNA and nrDNA sequences.**

| Source of variation | df | Sum of squares | Variance components | Percentage of variation | Fixation indices |
|---|---|---|---|---|---|
| cpDNA (all populations) | | | | | |
| Among populations | 1 | 53.821 | 1.19665[***] | 79.19 | $F_{ST} = 0.79195$ |
| Within populations | 155 | 48.727 | 0.31437[***] | 20.81 | |
| cpDNA(species) | | | | | |
| Among species | 1 | 53.821 | 1.17509[***] | 78.25 | $F_{CT} = 0.78248$ |
| Among populations within species | 23 | 22.242 | 0.12601[***] | 8.39 | $F_{ST} = 0.86639$ |
| Within populations | 132 | 26.485 | 0.20064[***] | 13.36 | $F_{SC} = 0.38576$ |
| cpDNA (V. rotundifolia) | | | | | |
| Among populations | 20 | 7.649 | 0.03763[***] | 19.64 | $F_{ST} = 0.19640$ |
| Within populations | 109 | 16.782 | 0.15396[***] | 80.36 | |
| cpDNA (V. trifolia) | | | | | |
| Among populations | 3 | 14.593 | 0.72257[***] | 63.14 | $F_{ST} = 0.63137$ |
| Within populations | 23 | 9.703 | 0.42188[***] | 36.86 | |
| ITS (all populations) | | | | | |
| Among populations | 1 | 1.075 | 0.01802 | 7.39 | $F_{ST} = 0.07392$ |
| Within populations | 175 | 39.501 | 0.22572 | 92.61 | |
| ITS (species) | | | | | |
| Among species | 1 | 1.075 | 0.01524 | 6.29 | $F_{CT} = 0.06286$ |
| Among populations within species | 25 | 7.854 | 0.01624 | 6.70 | $F_{ST} = 0.12983$ |
| Within populations | 150 | 31.647 | 0.21098 | 87.02 | $F_{SC} = 0.07146$ |
| ITS (V. rotundifolia) | | | | | |
| Among populations | 22 | 7.492 | 0.02231 | 10.09 | $F_{ST} = 0.10091$ |
| Within populations | 126 | 25.045 | 0.19877 | 89.91 | |
| ITS (V. trifolia) | | | | | |
| Among populations | 3 | 0.362 | −0.02429 | −9.69 | $F_{ST} = -0.09685$ |
| Within populations | 24 | 6.602 | 0.27509 | 109.69 | |

**Notes.**
Significance was tested by 1,000 random permutations.
[***]$P < 0.0001$.

**Table 5  Parameters of mismatch distribution Analysis of *V. rotundifolia* and *V. trifolia* based on cpDNA sequences.**

| | Tajima's *D* | P | Fu's *Fs* | P | SSD | P | $H_{RAG}$ | P |
|---|---|---|---|---|---|---|---|---|
| *V. trifolia* | 1.8690 | 0.8820 | 0.9575 | 0.7200 | 0.0051 | 0.5000 | 0.0350 | 0.7400 |
| *V. rotundifolia* | −0.3997 | 0.2830 | 0.2787 | 0.4830 | 0.0049 | 0.0000 | 0.1936 | 0.4800 |

Based on the cpDNA data, mismatch distribution analysis was performed to test whether the current expansion occurred in *V. rotundifolia* populations and *V. trifolia* populations separately. The results found that *V. rotundifolia* and *V. trifolia* populations each had a relatively smooth unimodal distribution, which was consistent with distribution under an exponential growth rate (Fig. 7), indicating that the two species had a rapid demographic expansion. In addition, among the 21 *V. rotundifolia* populations, Tajima's *D* and Fu's test gave apparently positive results, as evidenced by the value of *D* and *Fs* ($D = -0.3997$,

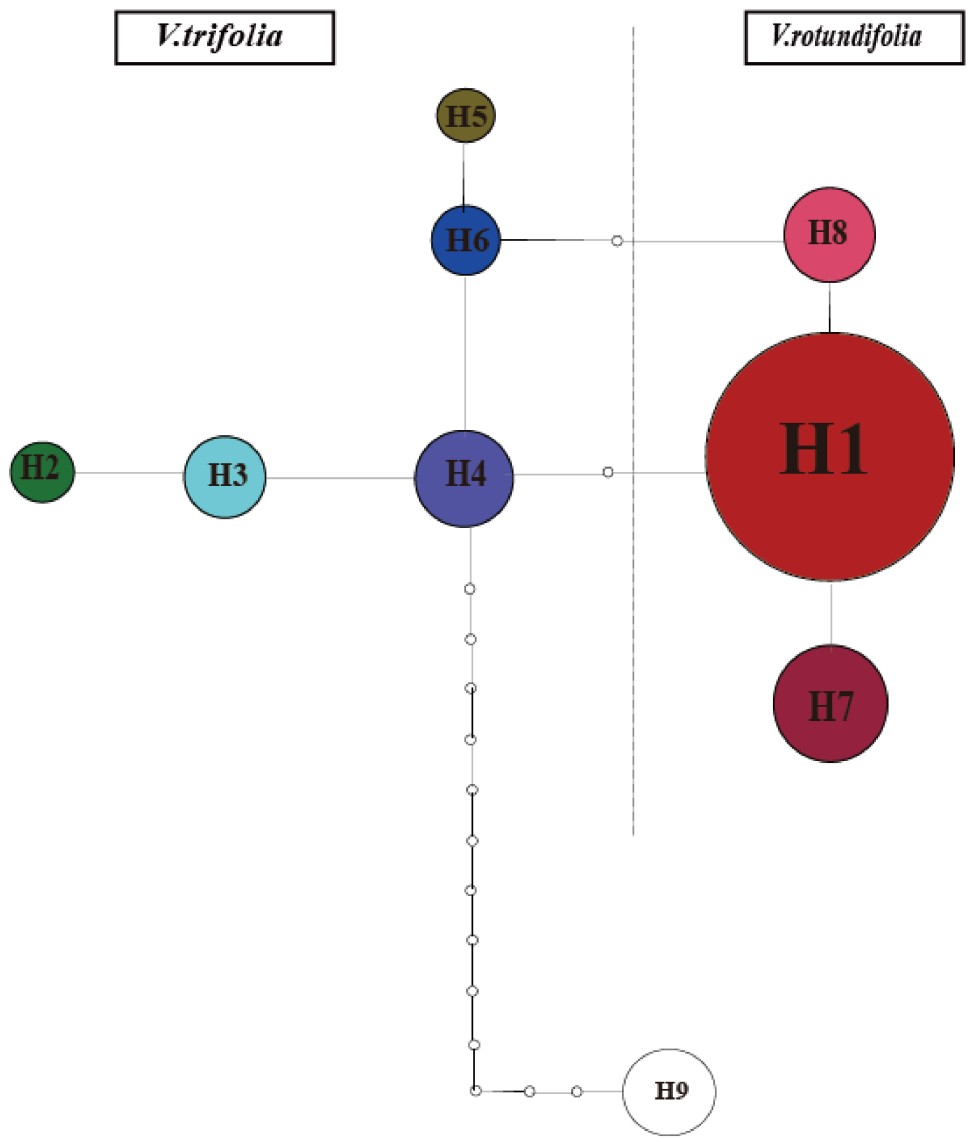

**Figure 4** **Minimum spanning network of eight cpDNA haplotypes in *V. rotundifolia* and *V. trifolia*.** The network was rooted at the *V. negundo*. Circle sizes are proportional to the number of samples per haplotypes. Hollow dots indicate the number of mutational steps.

*Fs* = 0.2787; both *P* >0.05), indicating that *V. rotundifolia* has not passed through a recent demographic expansion. In the four *V. trifolia* population, Tajima's *D* (*D* = 1.8690, *P* > 0.05) and Fu's *Fs* (*Fs* = 0.95759, *P* > 0.05) tests also showed no demographic expansion (Table 5). This result is inconsistent with that derived from the mismatch distribution plots. This discrepancy might be caused the insufficient number of variable sites provided by only two pairs of cpDNA regions (*trnG-trnS* and *psbA-trnH*).
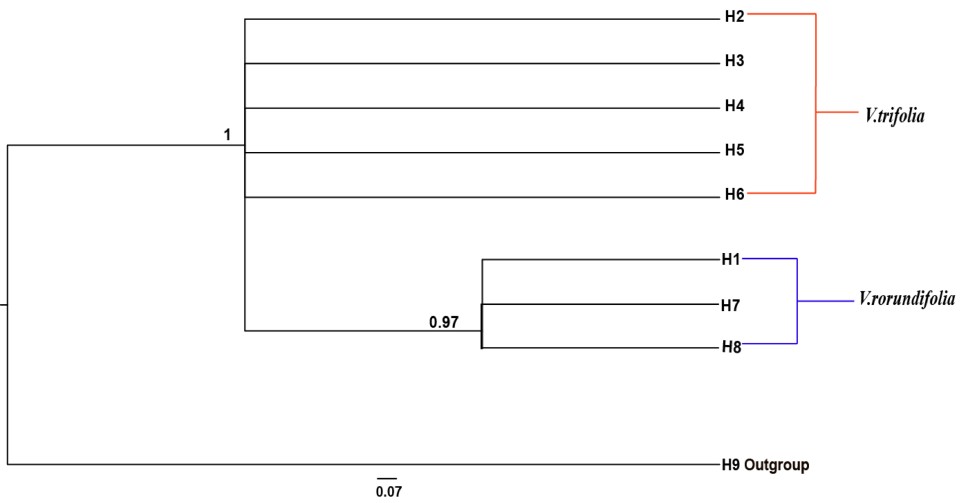

**Figure 5** **Bayesian inference (BI) phylograms of nine haplotypes based on chloroplast DNA (cpDNA) sequences.** The *V. negundo* (haplotype 9) was used as outgroup. Posterior probabilities (PP > 0.50) are labelled above the nodes.

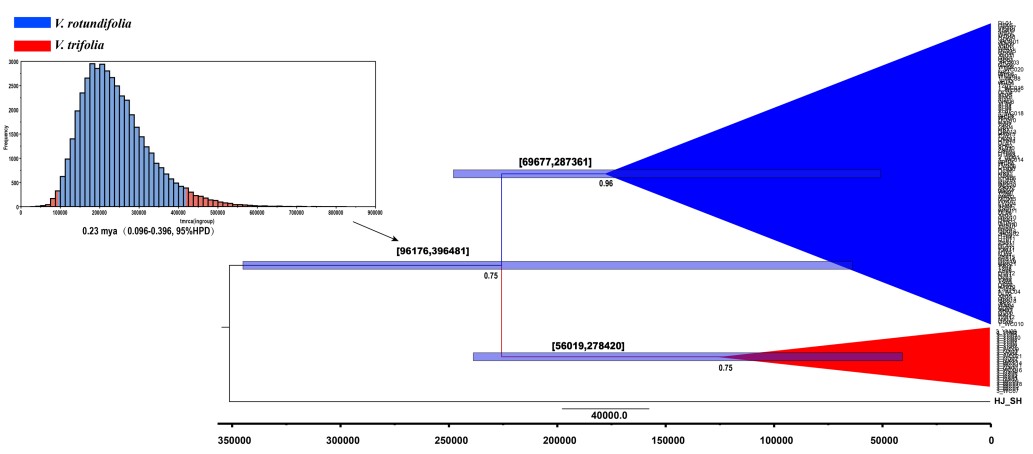

**Figure 6** **BEAST-derived chronograms of *V. rotundifolia* and *V. trifolia.* based on cpDNA sequences with *V. negundo* used as outgroup.** Posterior probabilities (PP > 0.50) are labelled below the nodes. The bars on each node indicate 95% highest posterior densities (HPDs) of time estimates (years ago).

## DISCUSSION

### A large-scale distribution and low genetic diversity of *V. rotundifolia*

*V. rotundifolia* is native to seashores from eastern India to Hawaii and from Japan to Australia, and can also be found in Brazil, Mauritius, Bangladesh, and Sri Lanka (*Moldenke, 1977*; *Munir, 1987*). In China, *V. rotundifolia* grows on the beach, seaside, and lakeside, mainly along the eastern coastal region (*Wang & He, 2013*). Based on our fieldwork, we investigated 14 sampling sites throughout the coastal areas and three sampling sites at lakesides in inland of China. The whole distribution areas of *V. rotundifolia* in China

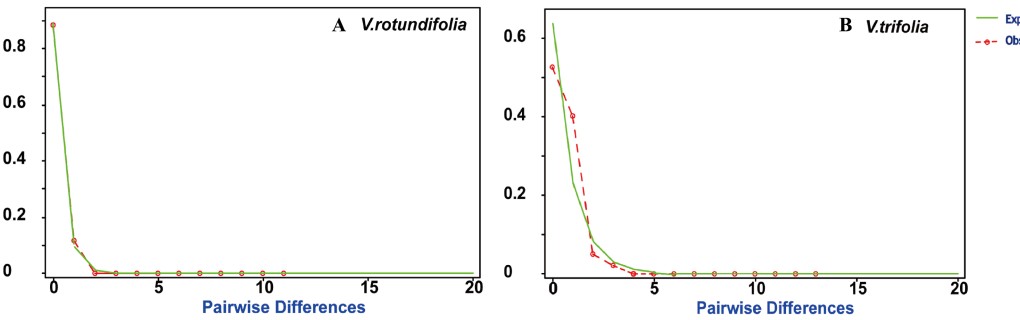

**Figure 7  Mismatch distribution plots for *V. rotundifolia* and *V. trifolia* haplotypes.** Observed (Obs) frequency of pairwise differences versus expected (Exp) under exponential population growth model for major cpDNA sequences. (A) *V. rotundifolia*; (B) *V. trifolia*.

were covered and also extended to Korea and Japan, including sites with few samples. The latitude and longitude of these locations range from $18°44'4.68$ to $39°51'26.24''$ and from $9°28'37.83''$ to $140°0'46.99''$, with the sampling distance crossing almost 3,000 km. However, the present study found that *V. rotundifolia* has extremely low levels of genetic diversity ($h_{d,cp} = 0.360$, $\pi_{d,cp} = 0.00010$; $h_{d,nr} = 0.440$, $\pi_{d,n_r} = 0.00000$) compared with most other plants (*Yan, 2007*; *Chen et al., 2008*; *Chen et al., 2012*; *Xing et al., 2017*). Such low level of nucleotide diversity has also been found in the coastal plant *Rhizophora* (Rhizophoraceae), with nucleotide diversities for *trnG-trnS* and *trnH-rpl2* being $0.04 \pm 0.02$ and $0.03 \pm 0.02$, respectively, and $\pi_{nr} = 0.02 \pm 0.01$ (*Lo, Duke & Sun, 2014*). However, for *V. rotundifolia*, using allozyme markers (*Yeeh et al., 1996*) and ISSR markers (*Hu et al., 2008*), moderate genetic diversity was reported, which was slightly higher than that observed in the present study. This discrepancy might be attributed to differences molecular markers used. In the present study, even when excluding poorly sampled populations, the secondary analysis also indicated that both the chloroplast and nuclear sequences have extremely low nucleotide diversity in *V. rotundifolia*.

The large-scale distribution of *V. rotundifolia* with a low level of genetic diversity may be attributed to three reasons: (1) Long-distance dispersal (LDD) by sea-drifted fruit: long-distance dispersal can result in the homogeneous distribution of DNA haplotypes at the spatial scale (*Miryeganeh, 2013*). Our results revealed that 17 of the 21 *V. rotundifolia* sampling sites, including the most distant populations covered by the geographical distribution range of sampling, such as XL (Xinglong of Hainan province) and JP (Japan), shared the common haplotype H1. This wide distribution of haplotypes may be caused by long-distance dispersal of this plant's fruit. The *V. rotundifolia* fruit is covered with a thick hydrophobic coating that can resist water penetration, rendering it able to float in the ocean, drift onto the beach, and disperse over long distance via sea drift (*Munir, 1987*; *Cousins et al., 2010*). Therefore, this rapid long-distance dispersal ability of *V. rotundifolia* fruit may be responsible for the low genetic diversity of this plant. (2) The similar ecological environment: life history traits and environmental factors affect the genetic diversity and structure of species (*Loveless & Hamrick, 1984*; *Nybom, 2004*). *V. rotundifolia* usually grows on beaches, sand dunes, and rocky shorelines at low elevations and can tolerate

the highly salt, drought, full sun, and sandy or well-drained soils (*Hawaiian Native Plant Propagation Database, 2018*; *GeoResources Institute (GRI), 2018*). These similar ecological niches could limit genetic differences and lead to a low genetic diversity of *V. rotundifolia*. (3) The asexual reproductive mode: this species proliferates by sexual reproduction, as well as clonal propagation through the elongation of rhizomes and root systems (*Gresham & Neal, 2004*). Clonal reproduction usually leads to the loss of genetic diversity within populations (*Cook, 1983*).

## Population structure and demographic history

Based on the cpDNA data, the results showed that all populations of *V. rotundifolia* and *V. trifolia* had a significant phylogeographical structure ($G_{ST} = 0.436$, $N_{ST} = 0.751$; $N_{ST} > G_{ST}$, $P < 0.05$). The current distribution of haplotypes may be caused by climate oscillation during the Quaternary period, which resulted in further large-scale migration of most plants and animals, and the subsequent accumulation of genetic variations and specific phylogeographical structures (*Hewitt, 2000*; *Hewitt, 2004*). *V. rotundifolia* populations showed an insignificant phylogeographical structure. ($G_{ST} = 0.201$, $N_{ST} = 0.462$; $N_{ST} > G_{ST}$, $P > 0.05$). Furthermore, AMOVA analysis showed low genetic differentiation among *V. rotundifolia* populations, indicating that rapid spread by sea currents may have played a key role. The unclearly phylogeographical structure and the low genetic differentiation in *V. rotundifolia* populations may relate to the rapid long-distance dispersal by sea drifts. This is not consistent with the report by *Hu et al. (2008)*, in which ISSR analysis showed a relatively high genetic differentiation ($G_{ST} = 0.587$) among populations of *V. rotundifolia*; the authors stated that such characteristics of *V. rotundifolia* are likely attributed to its sexual/asexual reproduction and limited gene flow. These contradictory results may have been caused by the different gene markers used (ISSR *vs*. cpDNA).

Estimated divergence times indicated that the genetic divergence between *V. rotundifolia* and *V. trifolia* occurred about 0.23 mya (0.096–0.396, 95% HPD), indicating a relatively short divergence time, corresponding to the late Pleistocene era, before the onset of the last glacial maximum (LGM). Following this divergence, long-distance dispersal may have been a major historical process for *V. rotundifolia*, because they are present throughout the Pacific, including the coastal areas of the continents and many islands. Additionally, we detected a recent population expansion of *V. rotundifolia* based on the apparent mismatch distribution, showing that *V. rotundifolia* had undergone a long-distance dispersal expansion in the short term.

The ancestral haplotypes are located in a central position within the phylogeographical network (*Crandall & Templeton, 1993*). Haplotype H4 is located in the center of the haplotype network, and is the closest haplotype to H9 (outgroup), and H1 has evolved from H4. Therefore, we suggest that H4 is the ancestral haplotype, which was detected in two populations (3_WC and 3_YN) in *V. trifolia*. Populations 3_WC and 3_YN are from the southern part of China and have higher haplotype diversities in all populations. Moreover, the results of the TCS network analysis also showed that the haplotypes shared by *V. rotundifolia* were deduced from those of *V. trifolia*. Thus, we speculated that Yunnan

and Wenchang regions might be two origin areas of *V. rotundifolia*. During the Quaternary period, *V. rotundifolia* expanded to eastern China rapidly.

The analysis of cpDNA data in this study showed that the *V. rotundifolia* and *V. trifolia* populations have no shared haplotypes. In addition, analysis of molecular variance (AMOVA) showed a significant difference in genetic differentiation between *V. rotundifolia* and *V. trifolia* ($F_{ST,cp} = 0.79195$). These findings supported the view that *V. rotundifolia* and *V. trifolia* are two separate species.

## Conservation strategy

It is essential to study the morphological variation, genetic diversity, and population structure to provide basic information on plant conservation (*Feng, Wang & Gong, 2014*; *Fan et al., 2017*). Climate change, the rapid growth of the human population, and economic development all contribute to the deterioration of *V. rotundifolia's* habitat, reducing the number of individuals in some populations. Therefore, *V. rotundifolia* resource protection is urgently required. The management priority is to conserve the population of the species that has the greatest diversity. Our results showed that the WN, XL, and QD populations have the highest haplotype diversity and nucleotide diversity in *V. rotundifolia*; therefore, conservation priority should be considered for these populations. *Hu et al. (2008)* proposed that the Xinjian population of *V. rotundifolia* from Jiangxi province of China had a considerably high genotypic diversity index and should be prioritized for protection measures.

Furthermore, population size is an important factor threatening the existence of species, and for many wildlife species, conservation should prioritize the smallest populations. Our field surveys found that most extant *V. rotundifolia* populations in China face a serious threat of extinction because of the limited number of individuals. Therefore, initial conservation measures for *V. rotundifolia* should be considered to increase the population size and genetic diversity. Among the extant populations, there are fewer than 10 individuals in the DL, LYG, and XL populations. Hence, we need to increase the population size for *in situ* conservation with high priority in these populations. In addition, habitat protection is also an important concern in *V. rotundifolia* conservation because of habitat damage by increased human activities and over-exploitation of the plant as a medical resource. Therefore, it is necessary to establish local nature reserves, especially for those small-sized and seriously disturbed populations. *Ex situ* conservation strategies, including germplasm collection, culture in botanical gardens (Kunming, Beijing), or reintroductions at appropriate areas with similar habitats, are also effective measures to preserve the genetic resources of *V. rotundifolia*. When the *ex situ* conservation is put in practice, samples should be collected from as many individuals as possible, because a large portion of the genetic diversity exists within, rather than among, populations. Sample collection from the WN, XL, and QD populations is recommended. Our experience, together with the results from previous studies, suggested that individuals in a population should be collected at >20 m spatial intervals to avoid collecting individuals with identical or similar genotypes (*Hu et al., 2008*).

## CONCLUSIONS

In summary, the present studies revealed the intraspecific genetic variation pattern of *V. rotundifolia* and *V. trifolia*, and identified genetic divergence that occurred during the late Pleistocene era. Our results showed that *V. rotundifolia* underwent a rapid long-distance dispersal expansion with low genetic differentiation among populations with an unclear phylogeographical structure. In addition, it has a low level of genetic diversity with a large-scale distribution. In conservation terms, the populations containing high genetic diversity (WN, XL, and QD) should be protected. A balanced and dynamic conservation strategy for *V. rotundifolia* has been proposed.

## ACKNOWLEDGEMENTS

We thank Yang Yang for providing samples for this study and for his initial help with lab work, Shunxing Zhang, Baorong Lu and Yanghaoyu (Jason) Chen for the English-language revisions.

### Funding

This study was financially supported by the National Natural Science Foundation of China (No. 81573599, No. 81371776), the Science and Technology Basic Work, Project of the Ministry of Science and Technology, China (No. 2015FY111500). The funders had no role in study design, data collection and analysis, decision to publish, or preparation of the manuscript.

### Grant Disclosures

The following grant information was disclosed by the authors:
National Natural Science Foundation of China: 81573599, 81371776.
Science and Technology Basic Work.
Project of the Ministry of Science and Technology, China: 2015FY111500.

### Competing Interests

The authors declare there are no competing interests.

### Author Contributions

- Yiqi Sun performed the experiments, analyzed the data, prepared figures and/or tables, authored or reviewed drafts of the paper, approved the final draft.
- Hong Yang performed the experiments, analyzed the data, contributed reagents/materials/analysis tools, authored or reviewed drafts of the paper, approved the final draft.
- Qiaoyan Zhang authored or reviewed drafts of the paper, approved the final draft.
- Luping Qin and Tingguo Kang conceived and designed the experiments, approved the final draft.
- Pan Li and Joongku Lee contributed reagents/materials/analysis tools, approved the final draft.

- Shichao Chen conceived and designed the experiments, analyzed the data, improved manuscript, replied reviewer and coordination team, approved the final draft.
- Khalid Rahman approved the final draft, English revision.
- Min Jia contributed reagents/materials/analysis tools, approved the final draft, English revision.

## Data Availability

   The sequence data are accessible via Genbank: MG822129–MG822626.

## Supplemental Information

Supplemental information for this article can be found online at http://dx.doi.org/10.7717/peerj.6194#supplemental-information.

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
