# Peer review of "Genetic diversity and its conservation implications of Vitex rotundifolia (Lamiaceae) populations in East Asia"

_PeerJ, doi:10.7717/peerj.6194_

## Round 0.1 · original submission · Major Revisions

Dear authors

Your ms has been reviewed. Our reviewers discovered many parts of your ms which need a thorough revision and even more analyses in the lab. Your revision, which will be sent to the same reviewers, can only be accepted if you follow their recommendations.

Kind regards
Michael Wink

Reviewer 1 ·

Basic reporting

see General comments for the author

Experimental design

see General comments for the author

Validity of the findings

see General comments for the author

Additional comments

Comments:

The ms. of Sun et al. describe the genetic variation of Vitex rotundifolia and V. trifolia in China, Korea and Japan.
The paper is principally well structured and done. Still, some points should be considered before publication.

A major point is the sampling. In many cases only 1 to 3 samples were analysed. This may cause a bias in the interpretation of the data. Low genetic diversity may be due to low sampling. The authors should critically address this problem. I understand that the species is rare, but it should be checked if the sampling affects the conclusions.

The calculation of the molecular clock is not appropriate. Wolfe et al. 1987 is too old. There are much younger papers like Kay et al. (2006). The authors should search for a suitable substitution rate close to Lamiaceae. Otherwise they can run BEAST without substitution rate prior. What is the conclusion or relevance of the outcome of this dating analysis? Pleistocene origin is very likely a priori.

The discussion on the potential refugia should be limited to the studied area.
Due to long distance dispersal this may be redundant.

The taxonomic conclusion is justified based on the presented data. However, to see if both species are really distinct one should include more samples of rotundifolia from other regions than E Asia. (To check if trifolia is nested in rotundifolia.)

The work by Hu et al. 2008 should be more incorporated in the discussion.

Details:

Legend fig. 6 chronograrms => chronograms; indicate 95% highest…
Fig. 6: Why is trifolia not reaching the present time (right)? It should align to rotund. May be a figtree problem.

l. 58 Drummond AJ et al; is wrongly cited.
l. 64 variety => variation
l. 73 only 2 papers dealing with controversial taxonomy of these 2 taxa?
l. 99 1-12 individuals in each population. One is not sufficient.
l. 111 Liter in small letters (l) for PCR
l. 274/5 What is the relevance of this sentence? Which context?
l. 278 . , delete
l. 292/6 rephrase 2 sentences. Difficult to understand.
l. 297 dispersed => distributed
l. 341 trifolia L & Littoralis Steenis

Reviewer 2 ·

Basic reporting

no comment

Experimental design

no comment

Validity of the findings

no comment

Additional comments

Sun et al. used cp DNA and nrITS to study the genetic diversity and structure of two species in Vitex (Lamiaceae). The authors sampled 27 populations and 177 individuals in total. Base on the genetic diversity, divergence time and demographic analysis, they found the two species show genetic differentiation, and low genetic diversity, and may diverge in the Pleistocene, afterwards underwent a potential rapid demographic expansion.

The study provides some new information of the genetic diversity for the populations of the two species in East Asia, and which further can be interpreted implications for practical conservation. However, they choose a regional sampling of the two species to test the demographical history and further try to clarify the delimitation of the two species, which I think it is not very appropriate. I suggest the authors focus on the genetic diversity, structure and its conservation implications of their data.

As the authors motioned in the introduction and discussion part of the two species, one of them are widely distributed in the Pacific coast, and the other is in Indian ocean coast. However, they only collected 27 populations from East Asia to test the species boundary and the demographical history. I do not think such data can reveal the true demographical history and clarify the species delimitation of the two species. Therefore, I suggest the author reorganize the MS.

In all, I think the authors need to pay much attention to the focus of the study and other details of the MS. Thus I will not suggest accepting to publish this MS at the current status.

Detail comments
Title:
I do not think the title can reflect the content of the MS. I suggest to change it like this: Genetic diversity and its conservation implications of Vitex rotundifolia (Lamiaceae) populations in East Asia

Abstract:
The abstract need to further refine according to my other my other suggestions.

Introduction:
The logic is not very well, this part should keep in logic consistent with the other section (such as material and method, discussion part). The last two sentences of the first paragraph are most abrupt.
The second paragraph is also not well organized

I think this part should be rewritten according to the questions they want to discuss.

Materials and Methods:
Four of the sampling site only consist of one individual each. I do not think we can call it population.

How many populations and individuals were sampled should be clearly stated here.

For molecular dating, why the authors used 1.0-3.0 × 10-9 substitutions per neutral site per year (s/s/y) (Wolfe, Li & Sharp, 1987) should be explained.

For demographic history, I do not think they need to do the test for V. rotundifolia and V. trifolia as a whole.

Results:

I cannot see the figure legend. If the Fig. 7 is right for the content in the MS?
The authors used a range of the mutation rate (1.0-3.0 × 10-9), but only one value (0.18 Ma) was shown here.

Discussion
I suggest the author focus on the genetic diversity, structure and conservation implications of the two species. Not too much for the demographical history and taxonomic implications.

Reference:
Some of the references styles are not consistent, which need to be carefully checked and corrected.

---

## Round 0.2 · Minor Revisions

Dear authors

Minor revisions are still required - please see the comments from Reviewer 1.

Best regards

Michael Wink
Academic editor

Reviewer 1 ·

Basic reporting

This is a review on a revised version of the paper by Sun et al. describing the genetic variation of Vitex rotundifolia and V. trifolia in China, Korea and Japan. The ms. is now better than the 1st version. Still, minor changes should be implemented.

Main point: The exclusion of poorly sampled populations should be mentioned in MM and results and shortly discussed in the discussion.

I have marked lots of smaller remarks and changes in the text.

Intro:
‚and also lacks analysis of the genetic variation background of this species‘ explain! The other studies did provide gen. variation!

Results:

0.23 mya (0.94-4.19 HPD) Something worng, probably with the lower HPD, when it is higher than average.


The English should be carefully checked by a native speaker!

Experimental design

see Basic reporting

Validity of the findings

see Basic reporting

Additional comments

see Basic reporting

Annotated reviews are not available for download in order to protect the identity of reviewers who chose to remain anonymous.

---

## Round 0.3 · accepted · Accept

Dear authors

Many thanks for the revision. As it is adequate, your ms can now be accepted.

Kind regards,

Michael Wink
Academic Editor

#